# Impact of the Interband Transitions in Gold and Silver on the Dynamics of Propagating and Localized Surface Plasmons

**DOI:** 10.3390/nano10071411

**Published:** 2020-07-19

**Authors:** Krystyna Kolwas, Anastasiya Derkachova

**Affiliations:** Institute of Physics, Polish Academy of Sciences, Al. Lotnikow 32/46, 02-668 Warsaw, Poland; Anastasiya.Derkachova@ifpan.edu.pl

**Keywords:** Localized Surface Plasmons (LSP), Surface Plasmon Polaritons (SPP), metal-dielectric interfaces, plasmon damping, plasmon resonance, dielectric function, dispersion relation, interband transitions, gold, gold nanoparticles, silver, silver nanoparticles, size effects, anomalous dispersion, quality factor

## Abstract

Understanding and modeling of a surface-plasmon phenomenon on lossy metals interfaces based on simplified models of dielectric function lead to problems when confronted with reality. For a realistic description of lossy metals, such as gold and silver, in the optical range of the electromagnetic spectrum and in the adjacent spectral ranges it is necessary to account not only for ohmic losses but also for the radiative losses resulting from the frequency-dependent interband transitions. We give a detailed analysis of Surface Plasmon Polaritons (SPPs) and Localized Surface Plasmons (LPSs) supported by such realistic metal/dielectric interfaces based on the dispersion relations both for flat and spherical gold and silver interfaces in the extended frequency and nanoparticle size ranges. The study reveals the region of anomalous dispersion for a silver flat interface in the near UV spectral range and high-quality factors for larger nanoparticles. We show that the frequency-dependent interband transition accounted in the dielectric function in a way allowing reproducing well the experimentally measured indexes of refraction does exert the pronounced impact not only on the properties of SPP and LSP for gold interfaces but also, with the weaker but not negligible impact, on the corresponding silver interfaces in the optical ranges and the adjacent spectral ranges.

## 1. Introduction

Theoretical and experimental studies of surface plasmons and plasmon-active surfaces have revealed the great potential of plasmonics basing on the exploration of properties of metal-dielectric interfaces. Surface plasmons are defined as collective electron oscillations coupled to the electromagnetic (EM) fields confined to the dielectric–metal interface. Strong local EM field enhancement and concentration of light energy into nanometer-sized volumes, opened up exciting prospects in sensing, detection, imaging and manipulation techniques at the nanoscale, as well as enabled the realization of various plasmonic devices and applications ranging from photonics, chemistry, medicine, bioscience, energy harvesting and communication to information processing (e.g., References [1,2,3,4,5,6,7,8,9,10,11,12,13,14,15,16,17,18,19]) and quantum optics (e.g., References [20,21,22,23,24,25,26,27,28,29,30,31]).

The solid-state approach to electronic excitations on a flat semi-infinite metal boundary resulted in understanding the very basis of surface plasmons phenomenon. The existing models of the many-body dynamical electronic response of solids were reviewed in Reference [32]. However, the intense developments of various plasmonic materials and metamaterials resulted in an increasing interest in their optical properties which are consequences of how they reflect, transmit, and absorb EM radiation. Electromagnetic characteristics of noble-metal based plasmonic structures and nanostructures, such as thin-films, hole-arrays or nanoparticles are known to strongly differ from their bulk counterparts, therefore stimulating efforts in attempts to understand the intrinsic dissimilarity and in the prediction of novel phenomena which manifests in observations and measurements.

Plasmonics is based on the excitation of surface plasmons. The surface plasmon phenomenon is connected with excitation of long-distance correlated oscillation of free electrons from the conduction band coupled to a transverse magnetic (TM)-polarised EM field at the metal-dielectric interface. Excitation of such surface oscillations forms the propagating waves called Surface Plasmon-Polariton (SSP) in the case of flat metal-dielectric interfaces, or the standing waves of Localised Surface Plasmons (LSP) in the case of (spherical) nanoparticles. Surface plasmon EM fields peak at metal-dielectric interfaces and decay exponentially away with distance from the interface. Losses and decoherence in the system of collectively oscillating electrons are caused by the scattering of electrons on background ions, phonons, and themselves (ohmic loss). Such collisional losses are augmented by the scattering on the confining surface which effectively shortens the mean free-electrons path in the case of smallest nanostructures.

In many applications, not the electron behavior in a plasmonic nanostructure is of prime interest but the EM fields coupled to charge oscillations and confined to the metal-dielectric interface. Such fields directly impact many processes that have found or are expected to be of potential in applications using propagating SPPs or those forming the standing waves of LSPs. Prediction for confined fields supported by the interface and their spatial distribution and dynamics are based on the self-consistent Maxwell’s electrodynamics in absence of the illuminating fields. In such an approach, field confinement arises from the properties of the metal-dielectric interface itself and is manifested when the interface is properly illuminated.

The reliable dielectric function (DF) which describes the manner the EM field acts on electrons of the materials involved, is the input parameter of EM theory. Significance of realistic description of dielectric properties of noble metals in basic issues and applications has been a motivation to many experimental studies intended to increase the accuracy of measurements of the indexes of refraction of noble metals in the function of wavelength [33,34,35,36,37,38]. In studying many aspects of plasmonic problems, it is convenient to have at hand an analytic expression for the dielectric function ε(ω) applicable for real metals, as it reveals the role of the useful parameters which characterize bulk metals. Such parameters are useful in predicting and shaping the plasmonic characteristics of the corresponding nanostructures. From this point of view, the realistic model of the frequency dependence of DF involving physically meaningful parameters and possibly small number of free parameters is of basic importance. The complex DF which reproduces well the measured real and imaginary parts of complex indexes of refraction of metals n(ω): $\varepsilon^{exp}\left(\omega \right)=n{\left(\omega \right)}^{2}={\left(n^{\prime }\left(\omega \right)+in^{\prime \prime }\left(\omega \right)\right)}^{2}$ is a fundamental quantity which is directly related to the electronic structure of metals. Such DF gives an insight into the elementary excitations of free carriers and interband absorption and allows us to study the basic surface plasmons characteristics supported by the interface. In particular, the model of DF for real metals must account for all important loss mechanism such as ohmic losses (heat release), inter-band absorption and scattering on surface. The additional damping can be caused by the limited physical dimension of the system resulting in the restriction of the mean free path of electrons. The phenomenological expression for the rate of this process called a “surface collision scattering rate” [39] is proportional to $v_{F}/a$ where $v_{F}$ is the Fermi velocity and *a* is the characteristic dimension of the metallic object. Let us note that such additional damping with the phenomenological “surface collision scattering rate” in nanoscale metals can be connected with Landau damping [40,41] and quantum confinement effect which takes place near metal-dielectric interfaces. However, in many studies, DF is modeled using a Drude-Lorentz approach which accounts for the intraband behavior of metals only. Such models, as well as those which approximately account for the presence of the interband transitions by a constant added to the real part of DF (e.g., References [42,43]), are known to be unsatisfactory in reproducing the experimentally measured indexes of refraction of noble metals. The frequency dependence of the interband transitions of metals is usually modeled by one or more Lorentzian-shaped contributions to DF what requires the use of large number parameters [39,44,45,46]. The optical properties of gold are more difficult to represent in the visible and near-UV region with an analytic model due to the more important role of interband transitions in these regions (e.g., Reference [39]).

In this paper, we critically reconsider the previously used Drude-type models of the dielectric function for gold and silver. Gold and silver nanostructures are most frequently used in either nanoscience or nanotechnology as they stand out due to superior plasmonic characteristics resulting from high optical conductivity and chemical inertness (especially gold) under ambient conditions. However, for gold, the models of the Drude-Sommerfeld-like dielectric functions often used in electrodynamic calculations are known to be fairly not perfect over the energy of 1.8 eV (e.g., References [42,43]) which is the lower threshold energy for the interband transitions (ITs). For silver, it is usually assumed that the Drude-type models of the dielectric function can be successfully used for the description of its optical and plasmonic properties, as the ITs fall in the ultraviolet. We reconsider this assumption and show to which extent such expectation can be valid. After taking into account the frequency-dependent contribution of the ITs in both metals we examine their role in tailoring SPPs on flat metal-dielectric interfaces and LSPs on nanoparticles versus their size. The SPP’s and LSP’s characteristic derived from the dispersion relations (DR) for flat and spherical interfaces form the complete tools for predicting the plasmonic performance of realistic but smooth gold and silver interfaces of basic architecture. Understanding the role of ITs in such elementary geometries of interfaces is of basing importance in predicting the plasmonic performance of many metal nanostructures with the potential application ranging from photonics, chemistry, medicine, bioscience, energy harvesting, communication and information processing.

## 2. Electronic and Optical Properties of Silver and Gold

The electron configurations of gold and silver atoms which belong to transition metals are exceptions to the Madelung (or Janet or Klechkowsky) rule which describes the filling order of the atomic sub-shells. These elements have completely filled d-subshells (respectively 4d${}^{10}$ and 5d${}^{10}$). Their metallic properties result from the single valence electron in the half-filled s-subshells (5s${}^{1}$ and 6s${}^{1}$ respectively). In spite of similar electron configurations, the optical properties (and color perceived) of gold and silver are different. In Ag the electronic transition responsible for absorption is the transition from 4d → 5s level, in Au it is 5d → 6s transition. However, in gold with the high atomic number (79), the electrons are subjected to an intense electrostatic attraction. Due to relativistic effects, the energy of the 5d orbital in Au is raised and the energy of the 6s orbital is lowered (e.g., References [43,47]) leading to the shift of absorption from ultraviolet to lower energies which fall in the blue visible range. As a result, gold appears yellow when illuminated with white light because the reflected light is lacking in blue. In silver the relativistic effects are smaller than those in heavier Au—the 4d-5s distance in Ag remains much greater; such transitions fall in the ultraviolet. Therefore, the color of silver appears to be silvery, like most other metals that result from the fairly flat reflectance spectra in the visible range. The absorption of light by gold and silver bulk metals takes place through intraband and interband transitions which occur in different energy ranges.

*Intraband transitions* (which occur at low energies) are free-carriers process. Electrons in the conduction band are considered as free electrons forming an electron gas and the long-range correlations in their positions is treated in terms of collective oscillations at the plasma frequency $\omega_{p}$ of the system as a whole. Intraband transitions result from excitations of electrons into a higher energy level within the same band. Intraband transitions contribute to the infrared absorption of free electrons as in the Drude model. In such model, incident EM radiation of frequencies below the plasma frequency $\omega_{p}$ is reflected, because the electrons in the metal screen the electric field of the light wave. EM radiation of frequencies above $\omega_{p}$ is transmitted, because the electrons cannot respond fast enough to screen it (e.g., References [48,49]).

*Interband transitions* are transitions from an occupied state below the Fermi level, to an unoccupied state in a higher band and are intrinsically a quantum mechanical process. In practice, only transitions to few energy bands contribute as in gold [50,51] where such transitions are known to be connected with the critical points (so-called Van Hove singularities) in the density of states which occur near symmetry points in the Brillouin zone. The large density of states in these regions is responsible for interband absorption (and emission) in the visible range which takes place from the top of the *d* band to states just above $E_{F}$ in the conduction band. These transitions occur with the threshold at $E_{L}$ = 2.4 eV in the visible range (below green light wavelength λ = 516.6 nm). The additional ITs in gold are due to the excitations of electrons from the 5d-band to unoccupied states in the 6sp-band above $E_{F}$ with the interband gap $E_{X}$ = 1.8 eV (red light wavelength λ = 688.8 nm). Both energies specific to the ITs in gold can be easily recognized in the dependence of the real and imaginary parts of the index of refraction *n* (according to Reference [34]) represented as a function of ℏω (dots in Figure 1). These energies are displayed as the distinct low energy threshold at $E_{X}$ = 1.8 eV in $n^{\prime }\left(\hbar \omega \right)$ dependence, and as a distinct disturbance in $n^{\prime \prime }\left(\hbar \omega \right)$ dependence over $E_{L}$ = 2.4 eV.

## 3. Models of the Dielectric Functions

In physical systems which obey causality and linearity condition, the real and imaginary parts of the dielectric function are related by the Kramers-Kronig relations. The real part of DF determines the polarization of the medium when it is subjected to an electric field while the imaginary part determines the amount of absorption. The usefulness of the dielectric function model in predicting the properties of real physical systems depends on how they are able to reproduce the measured refractive indexes in the interesting frequency (wavelength) range.

Our aim is to study the impact of the ITs in plasmonic issues in the most accurate manner possible using the most simplest algebraic forms with as little number of fitted parameters as possible. For that purpose we consider the three models of DF:the free-electron Drude-type DF $\varepsilon^{free}$:
(1)$$\begin{array}{c} \varepsilon^{free}\left(\omega \right)=1-\frac{\omega_{p}^{2}}{\omega^{2}+i\omega \gamma }, \end{array}$$
which describes the low frequency intraband contribution to the polarizability,the simplest effective DF $\varepsilon^{eff}$:
(2)$$\begin{array}{c} \varepsilon^{eff}\left(\omega \right)=\varepsilon_{\infty }-\frac{\omega_{p}^{*2}}{\omega^{2}+i\omega \gamma^{*}}, \end{array}$$
which accounts for contribution of the ITs in a form of a constant included in the real part of DF, andthe more realistic, broad-frequency-range DF $\varepsilon_{m}$:
(3)$$\begin{array}{c} \varepsilon_{m}\left(\omega \right)=\varepsilon^{free}\left(\omega \right)+\varepsilon^{ib}\left(\omega \right), \end{array}$$
which accounts for free-electron contribution and is frequency dependent in both the real and imaginary parts.

### 3.1. Free-Electron Drude-Type Dielectric Function

In many plasmonic problems, the DF have been simplified to a form given by Equation (Equation 1) that considers the dynamics of valence electrons in a metal as a collective harmonic oscillation of the electron charge density (e.g., References [42,53]), whose behavior is imprinted in classical polarizability.

The parameters $\omega_{p}$ and γ of the Drude free-electron DF (Equation (Equation 1)):(4)$$\begin{array}{c} \varepsilon^{free}\left(\omega \right)=1-\frac{\omega_{p}^{2}}{\omega^{2}+\gamma^{2}}+i\frac{\omega_{p}^{2}\gamma }{\omega (\omega^{2}+\gamma^{2})}, \end{array}$$
which we propose in this work for gold ($\hbar \omega_{p}$ = 8.6 eV and ℏγ = 0.072 eV) and silver ($\hbar \omega_{p}$ = 9.0 eV and ℏγ = 0.02 eV) allow to reproduce very well the low and high frequency range of $\varepsilon^{exp}\left(\omega \right)$ dependence in its real part (Figure 1c), red lines) and the low range frequency dependence in the imaginary part (Figure 1d), red lines) in gold and silver (left and right panels respectively). Accepted parameters for gold allow a good reproduction of Reεexp(ω) and $n^{\prime \prime }(\omega )$ in the frequency range up to about 1.1 eV and Imεexp(ω) and $n^{\prime }\left(\omega \right)$ up to about 1.8 eV (red lines, left panel of Figure 2). Therefore, the modeling of properties of plasmonic materials with the use of $\varepsilon^{free}$ in gold can be fully reliable in the IR range, which can be eventually extended up to 1.8 eV for quantities which are mainly defined by Imεexp(ω). For silver (see the right panel of Figure 2) the limiting spectral range of applicability of $\varepsilon^{free}$ ends at about 3.39 eV with little inaccuracy in the range from 1.7 eV up to 3.39 eV for Imεexp(ℏω) and $n^{\prime }\left(\hbar \omega \right)$ and about 2 eV for Reεexp(ℏω) and $n^{\prime }(\hbar \omega )$.

The proposed parameters, expressed in electronvolts for convenience, reproduce also very well the low-frequency range of real and imaginary part of the measured index of refraction, as demonstrated in Figure 1a,b, red lines. Therefore, the fitted parameters $\omega_{p}$ and γ are well fixed to the experimental data and are expected to possess the correct physical meaning. These are the plasma frequency $\omega_{p}={\left[n_{e}e^{2}/m^{*}\varepsilon_{0}\right]}^{1/2}$ and electron relaxation rate γ of the bulk metal accounting for all electron scattering processes which lead to ohmic losses and result in heat release. For γ≠0, electrons in the conduction band are not totally free but can move quasi-free.

#### Dielectric Function for Metals and Semi-Metals Versus Kramers-Kronig Relations

In plasmonics papers one can find a collection of different parameters used to define the Drude-like DFs $\varepsilon^{free}$ (and $\varepsilon^{eff}$) (e.g., References [35,54,55,56,57]) which are declared to reproduce well the Johnson and Christy [34] measurements. The problem of unambiguity in finding the physically meaningful parameters of such function which represent the corresponding real and imaginary parts of $\varepsilon^{exp}\left(\omega \right)$ result from the fact that these functions do not fulfill Kramers-Kronig relations (KKR) which connect the real and imaginary parts of many complex functions in physical systems that obey causality and linearity conditions.

The parameters resulting from the well behaved fit to the low frequency range of Imεexp(ℏω) (e.g., those using the relaxation time τ=(9.3±0.9)fs (τ=1/γ) from Reference [34]) for gold) can not be suitable to unambiguously determine the corresponding imaginary part and vice versa. The problem is illustrated in Figure 2. Blue dashed lines fit well to Imεexp(ω) but the same parameters do not allow to reproduce satisfactory Reεexp(ω) dependence.

The mathematical reason for this problem to occur lies in the fact that the functions $\varepsilon^{free}$ (Equation (Equation 1)) (and $\varepsilon^{eff}$ (Equation (Equation 2)) do not fulfil the condition of being analytic in the upper complex half-plane as they diverge with decreasing ω’s [58]. KKRs do not apply for such a class of functions. The resulting very basic problem of causality seems to be not fully resolved but it can be “patched” by—either reformulation of KKRs [59] which can include limitation of the range of integration interval of the KKRs equations [60] (this solution is often practiced), or by modification of the model of DF using quantum or classical approach [61,62]. For example, in the generalized Drude model of the classical approach, one lets γ to be complex and frequency-dependent. It turns out that the real part of γ then stays rather constant up to the plasma frequency and the imaginary part is of less importance [62].

### 3.2. Effective DF with Frequency Independent Contribution of the ITs

In many plasmonic studies the simplified model of Drude-like DF $\varepsilon^{eff}\left(\omega \right)$ (Equation (Equation 2)) with the effective parameters $\varepsilon_{\infty }$, $\omega_{p}^{*}$ and $\gamma^{*}$ is used. The ITs are accounted by the constant $\varepsilon_{\infty }-1$ added to Drude-like form of DF (Equation (Equation 1)).

The effective parameters which we accept for gold are [43,63,64]: $\varepsilon_{\infty }=9.84$, $\hbar \omega_{p}^{*}$ = 9.010 eV and $\hbar \gamma^{*}$ = 0.072 eV and for silver [63,64,65]: $\varepsilon_{\infty }=3.7$, $\hbar \omega_{p}$ = 8.9 eV and $\hbar \gamma^{*}$ = 0.02 eV.

The dielectric function Reεeff with such parameters changes the sign from negative to positive above: $\hbar \omega_{\mp }=2.87$ eV for gold. With these parameters, however, gold is transparent over this frequency which is not the case for the real metal. Accepted parameters for gold allow good reproduction of Reεexp(ℏω) and $n^{\prime \prime }(\hbar \omega )$ in the range up to about 2.9 eV and Imεexp(ℏω) and $n^{\prime }\left(\hbar \omega \right)$ up to about 1.8 eV (see left column of Figure 2a,b and Figure 1, green lines). Therefore, the modeling of properties of plasmonic materials with use of $\varepsilon^{eff}$ in gold can be reliable up to 1.8eV in the optical range. For silver (see the corresponding right columns in Figure 2) the limiting frequency of applicability of $\varepsilon^{eff}$ ends over 3.2 eV.

The advantage of using $\varepsilon^{eff}\left(\omega \right)$ is its simplicity, and in the frequency ranges below 1.8 eV in Au and 3.2 eV in Ag the modeling is expected to lead to reasonable predictions. We keep the model of DF $\varepsilon^{eff}\left(\omega \right)$ for further considerations in order to illustrate the spectral range of applicability for modelings surface plasmons and in order to demonstrate what are after-effects of using such function near and over these frequencies. We also intend to show the discrepancy of such results when compared with the more exact modeling accounting for the frequency dependence of the ITs. Keeping this model, we can also mimic the formal solutions of the dispersion relations of SPP over the frequency of $\omega_{\mp }$ where metal is transparent to EM radiation.

The parameters and the spectral ranges of applicability of DF functions discussed in this paper are collected in Table 1.

### 3.3. Frequency Dependence of the Its Reflected in DF for Realistically Described Gold and Silver

Realistic dielectric function $\varepsilon_{m}\left(\omega \right)$ (Equation (Equation 3)) is expected to represent all important electronic and optical features of real gold and silver and must include the contribution of the interband transitions, as discussed in Section 2. $\varepsilon_{m}\left(\omega \right)$ must represent correctly not only the real and imaginary parts of $\varepsilon^{exp}\left(\omega \right)$ dependence but also that of n(ω) in possibly largest range. Therefore, it must contain a frequency dependence of the ITs contribution εib(ω)=Reεib(ω)+Imεib(ω) which includes the appropriate threshold parameters (see Figure 2). This obvious statement in the optical range in case of gold will be farther extended to the higher frequency range including far ultraviolet in case of silver. In order to follow the frequency dependence of Imεexp(ω) in the higher frequency range (Figure 2), we propose the contribution Imεib(ω) to $\varepsilon^{ib}\left(\omega \right)$ in the form of the step-like logistic function:(5)Imεib(ℏω)=εlo+εup−εlo1+eS·ℏ(ωc−ω),
where $\varepsilon_{lo}$ and $\varepsilon_{up}$ are the lower and upper asymptotes, $\omega_{c}$ is the central frequency, and *S* is the slope of the step.

The function Imεib(ω) with the constant upper asymptote is suitable to reproduce well the step in Imεib(ℏω) up to about 4.7 eV (UV range) in gold (see the inset in Figure 2b), left column). The central frequency is $\hbar \omega_{c}=E_{L}=2.4$ eV, and the slope *S* must reproduce the lower-energy threshold of the ITs at $E_{X}=1.8$ eV. [43].

Figure 2a demonstrate that the correction to Reε(ω) would be also desirable around the characteristic frequencies for the interband transitions $\omega_{c}$ in both gold and silver metals. We introduce such frequency-dependent correction in harmonic-oscillator-based representations of the dielectric functions in the simplest possible form of the Lorentzian profile:(6)Reεib(ℏω)=Aℏ2(ω−ωc)2+(ℏγL)2,
with two fitted parameters *A* and $\gamma_{L}^{2}$. The many-parameter Lorentzian profiles have been previously used in the DFs models basing for example, on critical points analysis of metal band-structure (e.g., References [44,46]).

Let us note that in silver the interband transitions provoke the change of sign in Reεexp(ω) in the frequency range between $\hbar \omega_{\mp }\approx 3.9$ eV and $\hbar \omega_{\pm }\approx 4.7$ eV.

Table 1 includes all the predefined and fitted parameters of $\varepsilon_{m}\left(\omega \right)=\varepsilon^{free}\left(\omega \right)+\varepsilon^{ib}\left(\omega \right)$ with $\varepsilon^{free}\left(\omega \right)$ discussed and parametrised in Section 3.1. The free parameters of $\varepsilon^{ib}\left(\omega \right)$ result from the fit to the real and imaginary parts of $\varepsilon^{exp}\left(\omega \right)$ and n(ω). The resulting real and imaginary parts of $\varepsilon_{m}\left(\omega \right)$ and $\sqrt{\varepsilon_{m}\left(\omega \right)}$ dependencies for gold and silver are presented in Figure 1, (blue solid line) revealing reliability of $\varepsilon_{m}\left(\omega \right)$ up to about 4.2 eV in gold and to about at least 7 eV in silver. It is usually supposed that ITs have no influence on the plasmonic activity of SPs. Farther in the text we present the verification of such an assumption.

In the spectral ranges of validity extended to UV and far UV (in case of silver) ε(ω) represents all important characteristics of dielectric properties of real gold and silver which include the realistic contribution of the ITs. Reεm(ω) describes the strength of polarization induced by an external electric field, while Imε(ω) characterizes all the absorption and scattering losses (heat release and re-radiation) encountered when materials interact with EM field.

## 4. Characteristics of Surface Plasmons Supported by Metal-Dielectric Interfaces Derived from the Dispersion Relations

The most desired characteristics of SPPs and LSPs are shortened wavelength and enhanced field strength. The link between appealing spatial characteristics of surface plasmons results from the character of the dispersion relations which connect the allowed frequencies of plasmon oscillations to the wave vector (in the case of SPP) or to the (inverse) of the radius in the case of LSP supported by spherical metal nanoparticle (MNP) [53,63,65,66,67,68].

The dispersion relation, which is obtained from Maxwell equations in the absence of the illuminating radiation, and the parameters derived from, defines the inherent properties of the metal-dielectric interface. DR allows the formulation of the conditions enabling excitation of EM surface wave (SW) supported by the interface after the interface is properly illuminated. The solutions of Maxwell equations for EM field at the metal-dielectric interface which fulfill the appropriate boundary conditions (see e.g., References [42,69,70] the reviews), define the normal-mode solutions excitable on the interface. The homogeneous wave equations in both media are the starting point for the description; infinitesimally narrow metal-dielectric boundary, homogeneity, and charge neutrality of both media is assumed. The dielectric functions which describe material properties of the metal and dielectric are the external parameters of the theory. The scheme allowing to derive the necessary conditions to excite SPPs on a flat interface and LSPs on a spherical interface is sketched in Figure 3.

The dispersion relations for SPP plasmons propagating at flat metal-dielectric interfaces is usually solved for real ω’s and complex wave vectors k, while in the case of (spherical) metal nanoparticles the solutions with complex ω’s and real k are searched for.

The dispersion relations, in both cases of a flat metal-dielectric interface and a MNP embedded in the dielectric, corresponds to the poles of the scattering problem of the Fresnel coefficient (see e.g., Reference [69]) or scattering coefficients of Mie scattering theory [53,63,65,67,68].

### 4.1. Flat Metal-Dielectric Interfaces

The basic properties of propagating SPP supported by the flat metal-dielectric interfaces are usually illustrated in the simplest case of a single flat metal-dielectric interface separating metal and a dielectric with the complex $\varepsilon_{m}\left(\omega \right)$ and the real $\varepsilon_{d}=n_{d}^{2}>0$ correspondingly. In many basic approaches it is assumed that the metal is the idealized free-electron metal described by $\varepsilon_{m}\left(\omega \right)=\varepsilon^{free}\left(\omega \right)$ (Equation (Equation 1)). However, the usefulness of such modeling for real metals is limited, and understanding the basics of the phenomenon of plasmons based on such modeling can cause problems when confronted with reality.

The expected SPP properties are drastically dependent on the model of the dielectric function of the metal and on the radiative and nonradiative losses (included or not). In order to visualize the issue, we reconsider the dispersion relation and spatial profiles of SPPs for various models of DF discussed in Section 3. DR is derived by considering standard normal-mode solutions of the Maxwell equations with the appropriate continuity relations at the metal-dielectric interface with the use of the different models of the dielectric functions. A flat metal-dielectric interface, formed by a nonabsorbing dielectric in the half-space above *x*-axis (z>0) adjacent to a metal which fills up the half-space below *x*-axis (z<0) is considered as usual (see e.g., References [42,69,70] the reviews) (left panel of Figure 3). We seek the normal-mode TM oscillating solutions in the form of the plane wave propagating along the *x*-axis:(7)$$\begin{array}{c} e^{i(k_{j}r-\omega t)}=e^{ik_{⊥j}\left(\pm z\right)}\cdot e^{i(k_{\|}x-\omega t)}, \end{array}$$
with real ω’s and the vectors $k_{j}=\hat{x}k_{\|}+\hat{z}k_{⊥j}$ allowed to be complex. However, unlike the often studied case of simplified description of metals, we also study the case when not only Imk‖≠0 but also Rek⊥j≠0 (as follows from applying the DFs studied in Section 3). The resulting dispersion relations (see e.g., References [42,69,70]):(8)$$\begin{array}{c} k_{\|}\left(\omega \right)=\frac{\omega }{c}\sqrt{\frac{\varepsilon_{m}\varepsilon_{d}}{\varepsilon_{m}+\varepsilon_{d}}}, \end{array}$$(9)$$\begin{array}{c} k_{⊥j}\left(\omega \right)=\frac{\omega }{c}\frac{\varepsilon_{j}}{\sqrt{\varepsilon_{m}+\varepsilon_{d}}},j=m,d, \end{array}$$
form the basis for predictions of the spectral ranges in which the oscillating EM wave is confined to the metal-dielectric interface. $k_{\|}\left(\omega \right)$ and $k_{⊥j}\left(\omega \right)$ are complex components of the vector $k_{j}=\hat{x}k_{\|}+\hat{z}k_{⊥j}$ which are parallel and perpendicular to the interface. The imaginary parts Imk⊥j define the exponential decay of the plane wave (Equation (Equation 7)) in the plane perpendicular to the interface, so they decide about evanescent confinement to the interface or localization of the plane wave at the interface. Imk‖ defines the damping of oscillations along the interface. $k_{0}=\omega /c=2\pi /\lambda_{0}$ is the free-space wave vector. The wavelength of EM wave with the same frequency ω in the adjacent dielectric medium is $\lambda_{d}=2\pi c/\sqrt{\varepsilon_{d}}\omega$ and the wavelength of the wave propagating along the interface is λ‖=2π/Rek‖. The complex components $k_{\|}\left(\omega \right)$ and $k_{⊥j}\left(\omega \right)$ (Equations (Equation 8) and (Equation 9)) are connected by the relation:(10)$$\begin{array}{c} k_{⊥j}=\sqrt{\varepsilon_{j}k_{0}^{2}-k_{\|}^{2}}. \end{array}$$

In the following subsections we give the formal solutions for all the real and imaginary components of the $k_{j}\left(\omega \right)$ dependencies resulting from the four models of the dielectric functions $\varepsilon_{m}$ discussed above: $\varepsilon^{free}\left(\gamma =0\right)$, $\varepsilon^{free}$, $\varepsilon^{eff}$ and $\varepsilon_{m}$ (with *m* denoting Au and Ag for real gold and silver) in the frequency ranges embracing the frequency dependent ITs. We present solutions based on $\varepsilon^{free}\left(\gamma =0\right)\left(\omega \right)$ (black line), $\varepsilon^{free}\left(\omega \right)$ (red line), $\varepsilon^{eff}\left(\omega \right)$ (green line) and the most exact modeling based on $\varepsilon_{m}\left(\omega \right)$ which accounts not only for ohmic losses but also for the frequency dependence of ITs in the studied metal-dielectric interfaces (Section 3, Figure 1). We keep all the formal solutions in similar frequency ranges (even in the case when their physical meaning in some sub-ranges is dubious) in order to illustrate the discrepancies in the results for various quantities important in plasmonic with the aim to demonstrate what happens when such models are employed to understand basic SPP features by considering DR in spectral ranges where their applicability is dubious. The examples are $\varepsilon^{free}\left(\omega \right)$ or $\varepsilon^{eff}\left(\omega \right)$ which are not applicable in the high frequency range (Section 3.1 and Section 3.2) and derivation of the “resonance frequency” (as equal to $\omega_{p}/\sqrt{2}$), maximal wavelength contraction or appearance of anomalous dispersion region [42,71]) cannot be applicable to realistic metals.

In particular, we keep also the results for $\varepsilon^{eff}$ over the frequency $\hbar \omega_{\mp }$, as they mimic the solutions of the dispersion relation for metal-dielectric interfaces in the frequency range where the real part of DF for a metal change the sign and became positive, what means that at these frequencies the free-electrons are not able to screen EM field. In particular, we discuss also the results for the silver interface within exact modeling in the frequency range between $\hbar \omega_{\mp }$ and $\hbar \omega_{\pm }$ (Section 3), where ReεAg, where the screening by free-electron is overcome by the impact of IT.

#### 4.1.1. Ideal Free-Electron Metal—Perfect Localization

In order to meet the condition of the mode perfectly bounded to the surface (the most often discussed case, (e.g., References [42,69,70]), the normal component of $k_{j}$ in each medium must be imaginary, so each Rek⊥j=0. Such assumption applies to the idealized, lossless metal described by the real DF $\varepsilon \left(\omega \right)=\varepsilon^{free}\left(\omega ,\gamma =0\right)$ in the frequency range where $\varepsilon \left(\omega \right)<0,\varepsilon_{d}\left(\omega \right)>0$ and $|\varepsilon \left(\omega \right)|>\varepsilon_{d}$ (see Equation (Equation 8)).

If gold and silver were metals that meet these assumptions, the dispersion relations (Equations (Equation 8) and (Equation 9)) (and the resulting LSP properties) would be like those illustrated in Figure 4a (black solid lines for $\varepsilon^{free}\left(\omega ,\gamma =0\right)$ well hidden under the increasing part of the red line representing the expectations for $\varepsilon^{free}\left(\omega ,\gamma \neq 0\right)$.

We skip here the discussion of the dispersion relation for lossless metalic interface and the expected properties of SPP within such modeling, as it can be found in many books and review papers (e.g., References [42,69,72,73]).

#### 4.1.2. Lossy Metals

In lossy metals $k_{\|}$ and $k_{⊥j}$ components are complex and in general neither their imaginary nor real parts can be neglected *a priori*. The numerical calculations of k‖(ω)=Rek‖(ω)+iImk‖(ω) and k⊥j(ω)=Rek⊥j(ω)+iImk⊥j(ω) (Equations (Equation 8) and (Equation 9)) involve the square roots of imaginary numbers which, depending of the sign of the imaginary value under the roots, leads to four different solutions, in general. The final physical criterion applied in choosing the solutions which predict the damping in place of the amplification is:(11)Imk⊥d>0, Imk⊥m<0,(12)Imk‖>0,

The frequency dependence of those quantities is discussed in Section 4.1.2. The resulting SPP propagation length in relation to other length-scales characterising SPP wavelength and localization are discussed in Section 4.1.3.

##### Modification of the Dispersion Relation by the Interband Transitions

The most reliable (exact) modeling that accounts for the interband transition (blue lines in the following figures), introduces basic modifications to the dispersion relations of the surface-waves supported by the gold and silver interfaces in comparison with the simplified modeling.

The real parts of $k_{\|}\left(\omega \right)$ dependence (Equation (Equation 8)) which correspond to the condition (Equation 12) are presented in Figure 4a for all the considered models of DFs. In the lowest frequency range, Rek‖(ω) (blue solid line in Figure 4a) almost follows the light-line $\omega =k_{0}c/\sqrt{\varepsilon_{d}}$ (marked with dashed black line). At these frequencies the wave propagating along the interface is grazing the surface like in the simplified modeling (black, red and green solid lines). With increasing frequency, the dispersion relation Rek‖(ω) (blue lines in Figure 4) moves further away from the light-line. In case of silver, the distinct local maximum of Rek‖(ω) is reached. With still increasing frequency, DR displays a “turn-back” and crosses the light-line. Similar crossings can be observed also in free-electron metal (black and red lines) over the frequency equal or close to $\omega_{p}$ and for modeling using $\varepsilon^{eff}$ (see Figure 4a), green lines) over $\hbar \omega_{\mp }$ (2.8 eV for gold, and 4.1 eV for silver in very far ultraviolet). The important discrepancies of the exact modeling (blue lines) and those resulting from the simplified ones (black, red, and green lines) prove that near and over these frequencies, such simplified modeling is not applicable. In particular, the value $\omega /\sqrt{2}$ is often quoted as a frequency that corresponds to “the resonance frequency” of SPP, is obtained with the models of DFs (black and red lines) which do not apply in such high-frequency range as there are no real metals that can be described by those. Let us also note that the estimation of the maximal shortening of the SPP wavelength within modeling using $\varepsilon^{eff}$ in Reference [42] can not be reliable from the same reason.

However, Rek‖ is not a total wave-vector of the excitable mode at the interface, as Rek⊥j≠0. Also, |Imk⊥j(ω)| on both sides of the interface responsible for EM wave localization are strongly modified. Both introduce basic changes to the simplified understanding of LSP supported by real metal-dielectric interfaces.

The real parts of $k_{⊥j}\left(\omega \right)$ corresponding to the chosen solutions for Imk‖j(ω) (conditions (Equation 11)) are presented in Figure 5a,b. For all models of DF for lossy metals (red, green and blue line in Figure 5) Rek⊥j≠0.

The vector Rek⊥ of length:(13)Rek⊥(ω)=Rek⊥d(ω)+Rek⊥m(ω).
tilts the total wave vector Rek=Rek⊥+Rek‖ from the interface to the metal side (Rek⊥(ω)<0, see Figure 5c) spoiling the longitudinal character of the surface modes [69,74] and leading to the radiation leakage from the interface. The tilt angle of the total wave vector k of length:(14)k=Rek‖2+Rek⊥2
is defined by the arctan(Rek⊥/Rek‖).

##### Localization of SPP

The components Imk⊥d(ω) and Imk⊥m(ω) functions which are the measure of SPP localization, are illustrated in Figure 6a,b for all the studied models of the dielectric functions for gold (left column) and silver (right column) interfaces. The spectral range of relatively good localization of SW (small Imk⊥d,Imk⊥m) is dramatically reduced, when the interband transitions (green and blue lines) are accounted for in addition to ohmic losses included in $\varepsilon^{free}$ (red line) even in the smallest frequency range. The SPP mode is more tightly bound to the surface from the dielectric side. The silver interface allows much better localization than the gold one over full optical range (between 1.6 eV and 3.1 eV).

Figure 5c demonstrate that in gold (left column) and silver (right column) interfaces, the spectral ranges in which the longitudinal character of SW is only slightly affected, is dramatically reduced, when the ITs (green and blue lines) are accounted for in addition to ohmic losses (compare with the red line). The silver flat surface allows much better localization of the total energy near the interface over the full optical range (between 1.6 eV and 3.1 eV) than the gold one. In gold, the radiative leakage from the interface due to the ITs is present even in the smallest frequency range in IR spectral range (compare the blue and the red lines in Figure 5) and display fast increase with frequency. Let us note (see Figure 5) that in all models of DF for gold and silver which accounted for the ITs or/and ohmic losses (red, green and blue lines), the conditions Rek⊥(ω)=0 (and Rek⊥j(ω)=0) is never reached, so the modes supported by the realistic lossy gold and silver interfaces are never of purely longitudinal character.

##### Anomalous Dispersion of Surface Waves and Sub- and Superluminal Phase Velocities of SPP

Modification of the dispersion relations $k_{\|}\left(\omega \right)$ and k(ω) by the ITs is also reflected in the phase velocity $v_{\|}\left(\omega \right)$ (also in the energy propagation velocity not discussed in this paper) and in the wavelength $\lambda_{\|}\left(\omega \right)$ of the surface longitudinal wave propagating along the interface.

For the silver interface such modification takes place in a intriguing way revealing the well pronounced region of anomalous dispersion which manifests as the decrease in k(ω) and $k_{\|}\left(\omega \right)$ dependencies with increasing ω (Figure 7a,b, right column, blue lines). Such anomalous behaviour known from optics is accompanied by a dramatic and intriguing modification to the phase velocity v‖=ω/Rek‖ of SPP of the surface longitudinal wave supported by the silver interface (Figure 7c, blue line, right column), the fact just mentioned in Reference [75]. In the spectral range of the anomalous dispersion $v_{\|}\left(\omega \right)$ reaches the superluminal velocity (Figure 7c, blue line, right column). As is known from optics [76], in the frequency ranges of anomalous dispersion one can expect that the phase velocity or even group velocity of pure traveling wave can be greater than the speed of light. Moreover, the group velocity may exceed the speed of light in vacuum or become negative but it does not mean that such superluminal velocity corresponds to a velocity with which a signal is propagated [76]. Anomalous dispersion occurs only in regions of high absorption. In the discussed case of SPP on metal-dielectric interfaces, such high absorption is due to the interband transitions. Moreover, in the wavelength scales defined by $\lambda_{\|}$ (discussed in the next section), the oscillation length falls down (shadowed range in Figure 7c, right column) as we show in the next subsection.

#### 4.1.3. Lengthscales of SPP for Realistic Gold and Silver Interfaces

Inclusion of ITs into modeling redefines the lengthscales of the SPP mode supported by the interface. Farther we illustrate the corresponding quantities for SPPs, as they are of basic importance for SP-based photonics.

##### The Oscillation Length and the Propagation Length of SPP

Figure 8a illustrate the wavelength λ‖=2π/Rek‖ of the surface wave propagating along the interface found for realistic lossy metals (solid blue lines) and for free-electron metal (red solid lines). The corresponding long-dashed lines illustrate the LSP oscillation length $L_{\|}\left(\omega \right)$ defined as the propagation distance over which the field amplitude decays to 1/e of its original value: (15)L‖=1/Imk‖,
and the attenuation length $l_{\|}=L_{\|}/2$ (short-dashed lines) which is often used in the simplified modeling in which Rek‖=0.

The condition: $L_{\|}\left(\omega \right)\geq \lambda_{\|}\left(\omega \right)$ can be used to establish the spectral ranges in which the SPP oscillations exist; if $L_{\|}\left(\omega \right)<\lambda_{\|}\left(\omega \right)$, the amplitude of SPP oscillations decays faster than the oscillation lasts. Figure 8a and Figure 9a show that for gold interface, SPP oscillations can be excited in the full studied spectral range (at least to 4.2 eV) while in silver the oscillations are not excitable in the spectral range between 3.5 eV and 4.2 eV (the range of wavelengths $\lambda_{0}$ from 295 nm to 354.2 nm) which embraces the spectral range of anomalous dispersion in the UV spectral range.

Figure 8a and Figure 9a allow also to compare the SPP wavelength $\lambda_{\|}$ with the wavelength $\lambda_{0}=\lambda_{d}n$ (blue solid and black dashed lines respectively) at the same oscillation frequency ω (the same wavelength $\lambda_{0}$) and to access the spectral ranges of the maximal wavelength (and field intensity) compression. The SPP wavelength shortening $\lambda_{0}-\lambda_{\|}$ and the corresponding relative shortening $\left(\lambda_{0}-\lambda_{\|}\right)/\lambda_{0}$ are presented in Figure 8b and Figure 9b revealing existence of the maximal shortening at ℏω = 2.3 eV (about $\lambda_{d}$ = 540 nm) for gold and at ℏω = 3.6 eV ($\lambda_{d}$ = 344 nm) for silver interfaces. However, in the case of silver, the maximum falls in the spectral range of anomalous dispersion, where the short oscillation length $L_{\|}$ disables full oscillation cycle (Figure 9a), as $L_{\|}<\lambda_{\|}$ (the region shadow-marked).

Let us note that the relative wavelength contraction $\left(\lambda_{0}-\lambda_{\|}\right)/\lambda_{0}$ for gold is more effective that for silver interface (compare blue and black dash-dotted lines (in the left Figure 9b) in the almost whole visible range. $\left(\lambda_{0}-\lambda_{\|}\right)/\lambda_{0}$ can be treated as a measure for the electric field confinement of SPP.

Let us mention here that according to References [40,41] (and references therein) the increase of the electric field concentration produces two primary nonlocal effects: an increase in energy dissipation and an expansion of the region of SPP mode (diffusion). Electric field confinement enhances the loss due to Landau damping associated with the surface collision damping (an additional nonlocal effect resulting in the shortening of the free-electron path due to presence of the interface)) which effectively limits the degree of confinement.

##### Confinement (Localization) Length of SPPs

SPP plane wave is well bounded to the metal-dielectric interface when the wave amplitude decays exponentially in both the metal and the dielectric (Equation (Equation 7)) with the sufficiently large rate Imk⊥j. Figure 10 (blue lines) illustrate the confinement lengths $\delta_{j}\left(\lambda_{0}\right)$ of the SPP wave amplitude in both the metal and the dielectric (Equation (Equation 7)):(16)δj=(Imk⊥j)−1,j=d,m.

In the dielectric medium adjacent to the metal, the confinement length $\delta_{d}\left(\lambda_{0}\right)$ of the EM wave is smaller than $\lambda_{0}$ (Figure 10b) falling in the evanescent-wave regime.

The diffusion length in the metal $\delta_{m}\left(\lambda_{0}\right)$ is smaller than $\delta_{d}\left(\lambda_{0}\right)$ by an order of magnitude (Figure 10a). In case of gold interface $\delta_{m}$ displays the maximum at about 490 nm, in case of silver interface $\delta_{m}\left(\lambda_{0}\right)$ monotonically decreases with $\lambda_{0}$ in the optical range. As demonstrated, the ITs influence the diffusion length in both gold and silver interfaces in the whole visible range.

Usage of simplified models of DF which do not account for the ITs (red lines in Figure 10), the SPP diffusion lengths $\delta_{m}$ are smaller than in the case when the ITs are accounted for (see the blue and olive lines in Figure 10a).

Let us note also that in real metals the character of the confinement length $\delta_{m}\left(\lambda_{0}\right)$ (blue solid lines in Figure 10a) only roughly (compare e.g., Reference [1]) follows the main spectral behaviour of the optical skin depth $\delta_{\alpha }\left(\lambda_{0}\right)$ in a metal [48] in the visible range (blue dashed lines in Figure 10a):(17)$$\begin{array}{c} \delta_{\alpha }=\frac{\lambda_{0}}{2\pi n^{\prime \prime }}=2/\alpha , \end{array}$$
$\delta_{\alpha }$ describes the exponential decay of the field amplitude of light wave propagating in a medium with the complex index of refraction (e.g., Reference [48]). $n^{\prime \prime }$ is the imaginary part of the index of refraction, α is the absorption coefficient which describes the exponential decay of the intensity (or power) of an electromagnetic wave inside a material (Beer-Lambert law). The difference is obvious: $\delta_{\alpha }$ depends on $n^{\prime \prime }$ (Figure 10a, blue line), while Imδm of the SPP surface wave depends also on the real part of the index of refraction of the metal (Figure 1) and changes with the dielectric properties of the adjacent medium.

### 4.2. Localised Surface Plasmons (LSPs)

Metal nanoparticles (MNPs) are the important class of plasmonic nanomaterials of the optical features dominated by the ability to resonate with the incoming EM radiation. In resonance, the incoming EM field is effectively captured at the curved metal-dielectric interface forming the standing surface cavity waves of wavelength scaling with the MNP’s dimensions, therefore being much shorter than the incoming radiation wavelength.

Spectral properties of Localized Surface Plasmons on the spherical interface of a single MNP are usually described by using Lorentz-Mie scattering formalism from the analysis of the scattering, absorption or extinction cross-sections spectra for chosen radius *R* of the NMP which are written in the form of a sum of an infinite series of the spherical multipole partial waves l=1,2,3,… (e.g., References [76,77,78,79,80]).

However, for fundamental reasons and keeping in mind applications, the desired parameters are size-dependent oscillation frequencies $\omega_{l}\left(R\right)$ and decay rates $\Gamma_{l}\left(R\right)$ (or times $\tau_{l}\left(R\right)=1/\Gamma_{l}\left(R\right)$) of the excited oscillations of LPS’s modes l=1,2,3,… supported by MNP. $\omega_{l}$ and $\Gamma_{l}$ are not the explicit parameters of Lorentz-Mie scattering theory.

The functions of NMP’s size $\omega_{l}\left(R\right)$ and $\Gamma_{l}\left(R\right)$ can be derived in absence of the incoming EM wave from considering the dispersion relation [31,43,63,64,65] according to the scheme presented in Figure 3 which is common for flat and spherical metal/dielectric interfaces. In the spherical system of coordinates, dispersion relation takes the form:(18)$$\begin{array}{c} \sqrt{\varepsilon_{in}}\xi_{l}^{\prime }\left(k_{out}R\right)\psi_{l}\left(k_{in}R\right)+\sqrt{\varepsilon_{out}}\xi_{l}\left(k_{out}R\right)\psi_{l}^{\prime }\left(k_{in}R\right)=0, \end{array}$$
and is composed of complex special functions of complex arguments and corresponds to the poles of the coefficient ${}^{TM}B_{l}$[76] in Mie scattering theory. $\varepsilon_{in}\left(\omega ,R\right)$ and $\varepsilon_{out}$ are DFs of the metal sphere and of the outer dielectric environment, respectively, where the dependence on *R* accounts for the additional damping of free-electron movement in MNP caused by the limited particle dimension. The resulting shortening of the mean free path of electrons is accounted in the relaxation rate γ by adding the term $Av_{F}/R$, where $v_{F}$ is the Fermi velocity and *A* is the proportionality constant [39]. We accept A=0.33 and $v_{F}=1.4\cdot 10^{6}$ m/s for gold and silver NPs [43,81,82]. $k_{in}\left(\omega ,R\right)=\sqrt{\varepsilon_{in}\left(\omega ,R\right)}\cdot \omega /c$, and $k_{out}\left(\omega \right)=\sqrt{\varepsilon_{out}}\cdot \omega /c$ are the wave vectors inside the sphere, and in the sphere surroundings, respectively and *c* is the speed of light. The complex $\psi_{l}\left(z\right),$$\xi_{l}\left(z\right)$ are Riccati-Bessel spherical functions (of complex arguments) which can be expressed by the Bessel $J_{l+1/2}\left(z\right)$, Hankel $H_{l+1/2}^{\left(1\right)}\left(z\right)$ and Neuman $N_{l+1/2}\left(z\right)$ cylindrical functions of the half order.

#### 4.2.1. LSP’s Resonance Frequencies and Damping Rates Versus MNP’s Radius

Conditions for existence of the solutions of DF (Equation (Equation 18)) are strongly dependent on the form of DFs for the particle and its surroundings and exist only for the complex ω’s. [31,43,63,64,65]. Such conditions found for consecutive radii *R* provide the explicit size dependence of $\omega_{l}\left(R\right)-i2\Gamma_{l}\left(R\right)$, where $\omega_{l}$ is the oscillation frequency and $\Gamma_{l}$ is the damping rate of the modes *l* (*l* = 1 for the dipole mode) of surface localized electromagnetic fields (SLEM). The size characteristics of such modes are presented in Figure 11 for l=1,2…7 for gold (left column) and silver (right panel). Red, olive and blue lines represent the results for consecutive models of DFs: $\varepsilon^{free}\left(\omega ,R\right)$, $\varepsilon^{eff}\left(\omega ,R\right)$ and $\varepsilon_{m}\left(\omega ,R\right)$ respectively.

Figure 11 show that $\omega_{l}\left(R\right)$ and $\Gamma_{l}\left(R\right)$ became insensitive to models of DFs with increasing size in the ranges of large and very large MNPs. Such ranges shift to higher $\omega_{l}$’s with increasing the multipolarity *l*.

Inclusion of interband transitions (accounted in $\varepsilon^{eff}\left(\omega ,R\right)$ and $\varepsilon_{m}\left(\omega ,R\right)$) possesses an essential impact on $\;omega_{l}\left(R\right)$ with the decreasing *R*. The experimentally measured resonance positions [43,83,84,85] (which are usually limited to the dipole (*l* = 1) plasmon resonance only) are more accurately predicted by a modeling using $\varepsilon_{m}\left(\omega \right)$. For example, in the experiment using gold nanoparticles with slight deviations from a perfectly spherical shape embedded in oil ($n_{out}$ = 1.5) [83], the resonance positions are 2.18 eV (568 nm) for the radius R=30 nm, what coincide very well with our predictions (see the insets in Figure 11a). For gold and silver nanoparticles in water ($n_{out}=$ 1.33) the predictions basing on $\varepsilon_{m}\left(\omega ,R\right)$ (not presented here) also fit well to the experimental results [43,84,85].

In case of $\Gamma_{l}\left(R\right)$ dependence in the same range of sizes (Figure 11b), the description of the interband transition by the effective parameters of $\varepsilon^{eff}\left(\omega ,R\right)$ in gold is not satisfactory. For $\varepsilon_{m}\left(\omega ,R\right)$, $\Gamma_{l}\left(R\right)$ is several times augmented in the smaller NP’s range when compared with the simplified models of DF for gold. In silver the effect is less pronounced. For smaller particles $\Gamma_{l}$’s display a farther increase due to the radius dependent surface damping (see insets in Figure 11b).

#### 4.2.2. Quality Factors of LSP Modes Versus MNPs Radius

The quality factor $Q_{l}\left(R\right)$ for the successive EM modes *l* supported by a spherical interface which we consider here is based on the definition of a quality factor of a resonator defined as the number of oscillations required for an energy of a freely oscillating system’s to fall off to $e^{-2\pi }$ of its original energy. The radius dependence of the quality factor $Q_{l}$ of a resonator formed by the spherical metal/dielectric interface of gold and silver MNPs
(19)$$\begin{array}{c} Q_{l}\left(R\right)=\frac{\tau_{l}\left(R\right)\omega_{l}\left(R\right)}{2}=\frac{\omega_{l}\left(R\right)}{2\Gamma_{l}\left(R\right)} \end{array}$$
is presented in Figure 12. Quality factors for silver NPs is several times better than for gold NPs of the same size up to very large NPs radii. Each $Q_{l}$ displays a maximum $Q_{l}^{max}$ which undergoes the shift with the increasing *l* toward larger sizes. Let us note that spherical metal/dielectric interface is not the best resonator for the dipole *l* = 1 mode (Figure 12b), as $Q_{l}^{max}$ grows with increasing multipolarity *l*.

Let us also note that for the regions of sizes where the quality factor in the dipole mode takes the maximal value $Q_{l=1}^{max}$, $Q_{l}$’s with l>1 are comparable or even higher (silver NPs) than those for l=1. With increasing size $Q_{l=1}\left(R\right)$ decreases. Therefore, one can expect that in MNPs of smaller sizes not only dipole but also higher multipolarity modes contribute to the energy storage at the interface (when the NP is illuminated), while with the increasing radius the contribution of the dipole mode (and successively lower-polarity modes) is eliminated.

#### 4.2.3. Cross-Section Spectra

The differences in $\omega_{l}\left(R\right)$ and $\Gamma_{l}\left(R\right)$ for various models of DF are clearly reflected in the spectral positions, number of spectrally resolved modes *l*, amplitudes and widths of the maxima displayed in the scattering and absorption cross-sections $C_{scat}$ and $C_{abs}$, calculated on the base of Lorentz-Mie scattering theory. Figure 13 demonstrate such spectra for the example radii of gold (left panel) and silver (right panel) MNPs. The axes are chosen in a way allowing to compare the relation of scattering and absorbing abilities of NPs of chosen size.

The positions of the maxima in the spectra (Figure 13) are strongly affected by ITs and the spectral resolution of the consecutive contributions of modes *l* = 1,2,3... is reduced. It is the consequence of the fact that the resonant frequencies $\omega_{l}$ with the corresponding *l* = 1,2,3,... are similar in the broad range of radii starting from small *R* (see Figure 11a) due to the flattening of $\omega_{l}\left(R\right)$ dependence.

The absorption remains dominant in smaller MNPs (the example of gold and silver MNPs with *R* = 5 nm, Figure 13a) but both scattering and absorption are strongly damped by the interband transitions (compare the data represented by blue line for $\varepsilon_{m}$ with those represented by red line $\varepsilon^{free}$).

In larger MNPs (Figure 13b), the example for gold and silver NPs of radius *R* = 30 nm), scattering abilities of NPs are augmented following the increase of the radiative contribution $\Gamma_{l}^{rad}\left(R\right)$ to the total rate $\Gamma_{l}\left(R\right)$ (Figure 11b), blue line), as discussed elsewhere [16,64].

With still larger MNPs (Figure 13c), the example for gold and silver NPs of radius *R* = 75 nm), the scattering cross-sections are still increased, due to the increase in $\Gamma_{l}^{rad}\left(R\right)$. Let us note that $\Gamma_{l=1}^{rad}$ reaches the near-maximal value for silver NP of this size (see Figure 11b), blue line).

The scattering and absorption abilities of gold and silver MNPs are summarised in 3D Figure 14. The efficiencies ( total cross-sections per unit surface) $Q_{scat}\left(\omega ,R\right)=C_{scat}\left(\omega ,R\right)/\pi R^{2}$ and $Q_{abs}\left(\omega ,R\right)=C_{abs}\left(\omega ,R\right)/\pi R^{2}$ are presented up to the size range ( to the radius *R* = 300 nm) which are still experimentally unavailable. Figures show, that the maximal optical performance of MNPs in (ω,R) plane is shifted toward larger nanospheres than those experimentally used. Figures demonstrate also the significant differences in the position of such maxima for scattering and absorption and between gold and silver MNPs.

## 5. Conclusions

Understanding and modeling of surface plasmon phenomena on lossy metals interfaces based on simplified models of DF causes problems when confronted with reality. A simplified description of plasmons (which assumes that DF is real and valid up to the high-frequency limit) not only leads to errors in the estimation of plasmonic parameters such as oscillation frequencies or damping rates of oscillations, but also makes the very understanding of this phenomenon inaccurate or faulty. The example can be the often met expectation for the SPP “resonance frequency” at $\varepsilon_{p}/\sqrt{2}$ derived from the high-frequency limit of the dispersion relation, in spite the free-electron dielectric function describes the properties of metals in the range of very low frequencies only. Another example is the incorrectly estimated localization length or wavelength shortening of SPPs. Similar discrepancies exist for the dipole plasmon resonance frequency in MNPs which appears for the frequency much lower than $\varepsilon_{p}/\sqrt{3}$ expected when the interband transitions are not accounted for. Nevertheless, $\varepsilon_{p}/\sqrt{2}$ and $\varepsilon_{p}/\sqrt{3}$ are often cited in review papers, lectures, and repeated in Ph.D. theses and dissertations.

Let us emphasize that the dispersion relations for surface plasmons basically characterize not the EM field which is absent but the inherent properties of the interface. The resulting self-frequencies and damping rates are parameters which can manifest in a form of (eventually shifted and broadened) resonances in the observed quantities when the nanostructure is properly illuminated. However, in general, such parameters can manifest in a different manner. The spectral position of the maxima, the amplitudes, and spectral widths can change with the quantity which is measured (e.g., References [86,87,88]) (see the example of LSPs manifestation for gold and silver MNPs of diverse sizes in Figure 13) and size of an MNP. We demonstrate that the quality factors, which are the measure of the energy stored in the oscillations of plasmon modes in MNP are larger for MNP of larger size and multipolarity. This effect is accompanied by the known effect of lower scattering and absorption losses in the corresponding size ranges.

In real lossy metals such as gold and silver, both the free-electron intraband and interband transitions are frequency dependent. The result of such transitions accounted in the phenomenological model of the dielectric function should be represented by the parameters allowing reproducing well both the experimentally measured complex indexes of refraction n(ω) and the resulting $\varepsilon^{exp}\left(\omega \right)=n{\left(\omega \right)}^{2}={\left(n^{\prime }\left(\omega \right)+in^{\prime \prime }\left(\omega \right)\right)}^{2}$. Inclusion of frequency-dependent radiative losses in addition to ohmic losses does exert the pronounced impact not only on the properties of SPP and LSP supported by gold interfaces but also, with the weaker but not negligible impact, on the corresponding characteristics of silver interfaces in the optical ranges and the adjacent spectral ranges.

The performance of plasmonic devices correlate with numerous parameters that need to be studied to reach the optimum and desired properties. The performed detailed analysis of all important surface plasmon parameters based on the dispersion relations both for flat and spherical gold and silver interfaces allows characterizing equivalently both geometries which are basic in the experimental practice and in applications. Discrepancy of the results for lossy metals (interband transitions included) and free-electron hypothetical metals reveals dramatic impact reflected in the expected properties of surface wave accepted by such metal-dielectric interfaces. Losses are an integral part of plasmonic systems and, as demonstrated above, can have a significant impact upon the physical understanding of surface plasmon phenomenon. In particular, consideration of the presence of frequency-dependent transitions in the extended frequency range revealed the presence of anomalous dispersion behavior for the silver flat interface in the near-UV, which, as far as we know, has not been experimentally studied.

The frequency-dependent interband transitions accounted in the model of the dielectric function exert the pronounced impact not only on the properties of SPP and LSP parameters for gold flat and spherical interfaces but also, with the weaker impact, for the corresponding silver interfaces in the optical range and the adjacent spectral ranges. Therefore, among many experimental studies which deal with SPPs in thin films, hybrid structures and nanowires (e.g., References [89,90,91]), the experimentally obtained dispersion relation bends to the higher frequencies (“to the right of the dispersion curve”), if compared with predictions based on simplified models of DFs. In particular, in a case of nanowires (e.g., Reference [89], the conclusion of slowing down the propagation velocity of SPP compared to the corresponding planar interface is not fully justified as the bending of the experimental dispersion curve can result from the presence of interband transitions which are not accounted in the modeling.

Inclusion of interband transitions (accounted in $\varepsilon^{eff}\left(\omega ,R\right)$ and $\varepsilon_{m}\left(\omega ,R\right)$) possesses also an essential impact on predicted self-energies $\omega_{l}\left(R\right)$ of spherical metal interfaces. The experimentally measured spectral positions of LSPs manifestation [43,83,84,85] (which are usually limited to the dipole plasmon resonance (*l* = 1) only) are very well predicted by the presented modeling using $\varepsilon_{m}\left(\omega \right)$ for $n_{out}$ = 1.5 (the case studied above) and, as we checked also for $n_{out}$ = 1.33 (i.e., for MNPs embedded in the immersing oil and water correspondingly).

Therefore, in order to understand quantitatively the dispersion, localization, and attenuation characteristics of surface plasmons over a wide range of wavelengths it is necessary to consider the metal as described by empirically determined indexes of refraction [92]. The analysis presented in this paper which is based on the form of the dielectric functions with the physically meaningful parameters (and able to reproduce the frequency dependence of such empirical indexes) allows analyzing the role of the individual parameters which characterize electronic and dielectric properties of metals. Such approach forms a common platform for describing plasmons localized on curved (spherical) and an extensively studied flat metal-dielectric interfaces.

## Figures and Tables

**Figure 1 nanomaterials-10-01411-f001:**
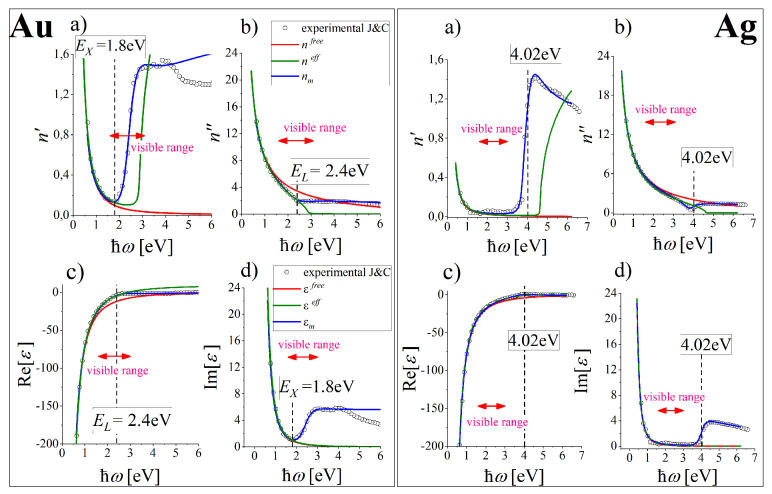
Subfigures (**a**) and (**b**), black circles: the measured real $n^{\prime }$ and imaginary parts $n^{\prime \prime }$ of index of refraction *n* for gold and silver [34] (Kramers-Kronig relations involved) and (**c**) and (**d**): the corresponding real and imaginary parts of the “experimental” function $\varepsilon^{exp}\left(\hbar \omega \right)={\left[n^{\prime }\left(\hbar \omega \right)+in^{\prime \prime }\left(\hbar \omega \right)\right]}^{2}$. Real and imaginary parts of
$\varepsilon^{free}\left(\hbar \omega \right)$, $\varepsilon^{eff}\left(\hbar \omega \right)$ and $\varepsilon_{m}\left(\hbar \omega \right)$ (Equations (Equation 1)–(Equation 3)) and the corresponding real and imaginary parts of the indexes of refraction fitted to both $\varepsilon^{exp}\left(\hbar \omega \right)$ and n(ω) are presented with red, green and blue lines correspondingly. Vertical lines mark the threshold energies for interband transitions for gold [50,52] and silver.

**Figure 2 nanomaterials-10-01411-f002:**
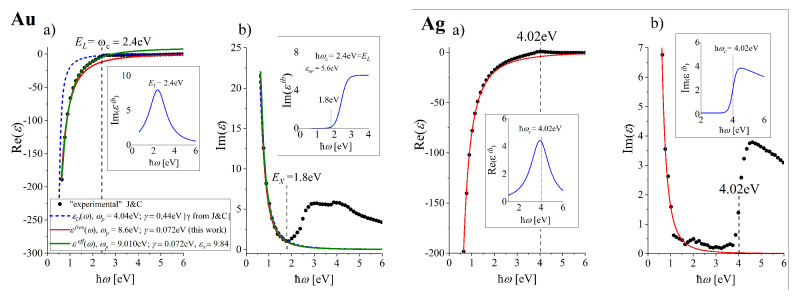
Illustration of the problems in choosing the reliable parameters for Reεfree(ω) in the low frequency range using the parameters of the imaginary part well fitted to low-frequency Imεexp(ω) dependence (**a**, blue dashed lines). The additives to $\varepsilon^{free}\left(\hbar \omega \right)$ (red lines) which describe frequency dependent interband transitions (Section 3.3) are presented in insets for gold and silver (**a**,**b**, respectively). The resulting $\varepsilon_{m}\left(\omega \right)$, Equation (Equation 3), reproduces well $\varepsilon^{exp}\left(\omega \right)$ and n(ω) in a broad spectral range (see blue lines in Figure 1).

**Figure 3 nanomaterials-10-01411-f003:**
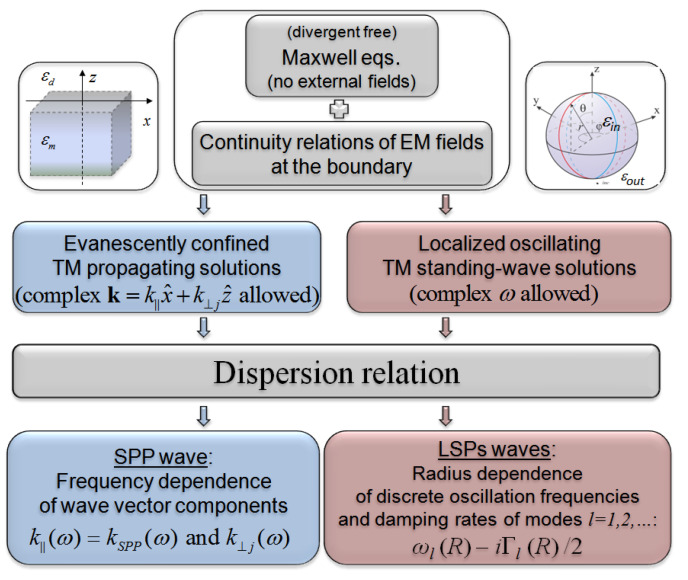
Schemes for deriving the dispersion relations for Surface Plasmon Polaritons (SPP and Localized Surface Plasmon (LSP) surface waves on a flat and spherical interfaces.

**Figure 4 nanomaterials-10-01411-f004:**
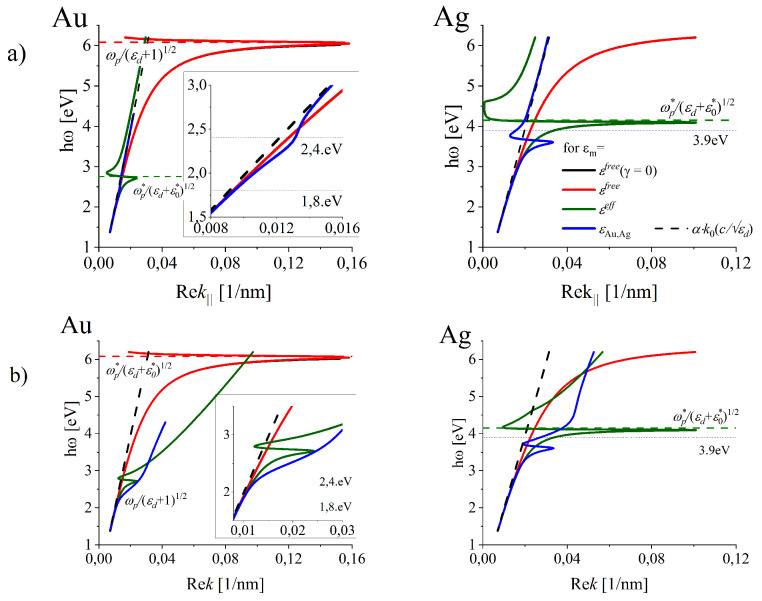
Dispersion relations for electromagnetic (EM) waves propagating along the metal-dielectric interface for metals of properties represented by various models of the dielectric functions. Blue lines represent the dispersion relations Rek‖(ω) (Equation (Equation 4)), (**a**) and Rek(ω) (Equation (Equation 14)) (**b**) found for $\varepsilon_{m}\left(\omega \right)$ which accounts for ohmic losses and frequency dependent interband transitions in gold (**left column**) and silver (**right column**).

**Figure 5 nanomaterials-10-01411-f005:**
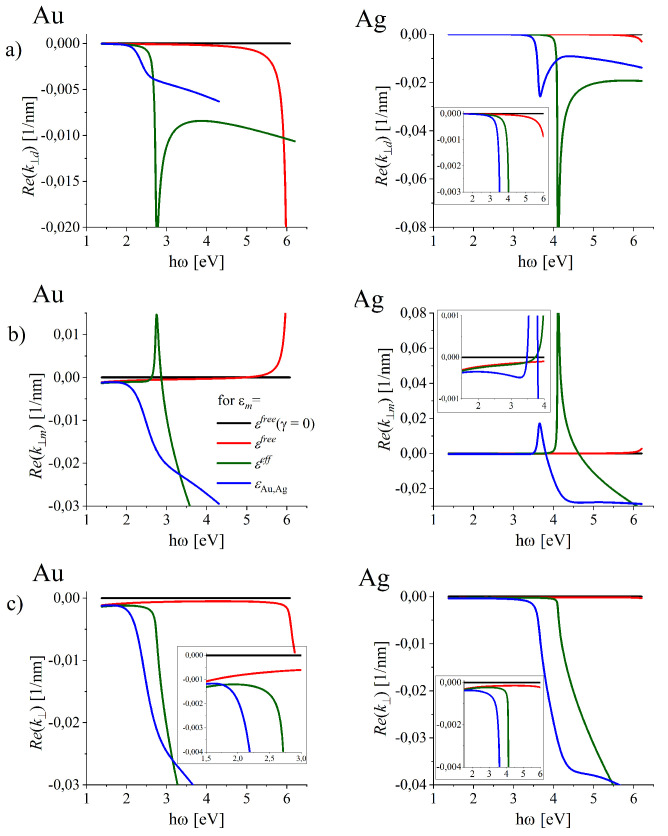
Radiation loss rates caused my the (**a**) dielectric and (**b**) metal and (**c**) the resulting Rek‖(ℏω) (Equations (Equation 13)) which is the measure of the total radiation losses of SPP for gold (left column) and silver (right column) of properties represented by various models of the dielectric functions. Blue line represents the most reliable localization rates found for $\varepsilon_{m}\left(\omega \right)$ accounting for ohmic losses and interband transitions.

**Figure 6 nanomaterials-10-01411-f006:**
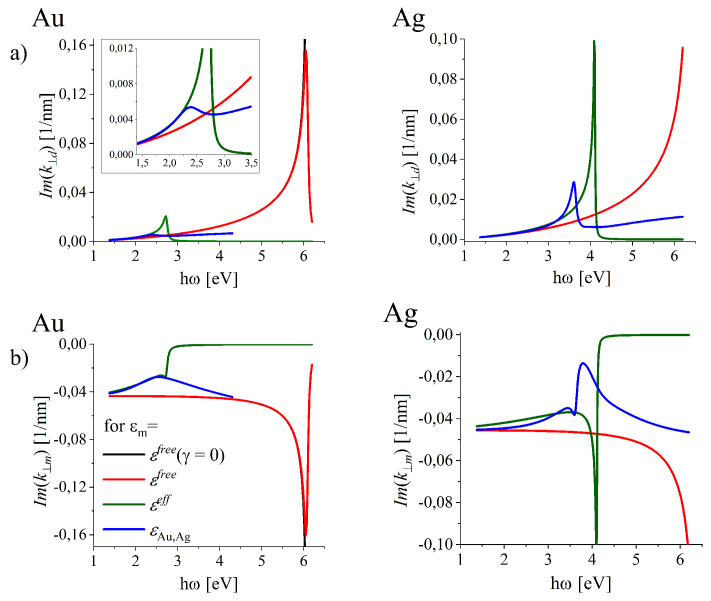
The rates Imk⊥d(ℏω) and Imk⊥m(ℏω) (Equations (Equation 11) and (Equation 12)) which are the measure of SPP localization from the dielectric and metal side of the interface for gold (**a**) and silver (**b**) of properties represented by various models of the dielectric functions. Blue line represents the most reliable localization rates found for $\varepsilon_{m}\left(\hbar \omega \right)$ accounting for ohmic losses and interband transitions in realistically described gold (**a**) and silver (**b**).

**Figure 7 nanomaterials-10-01411-f007:**
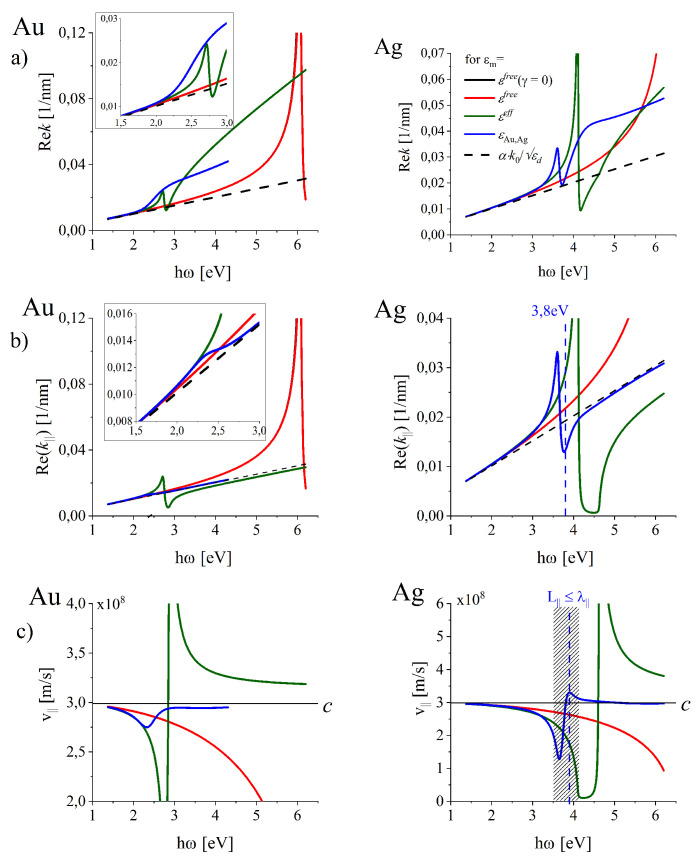
Frequency dependence (**a**) of the on-interface and (**b**) of the total wave vector of the SPP wave demonstrating the regions of the anomalous dispersion in Ag and (**c**) the resulting phase velocity of the on-the-interface wave.

**Figure 8 nanomaterials-10-01411-f008:**
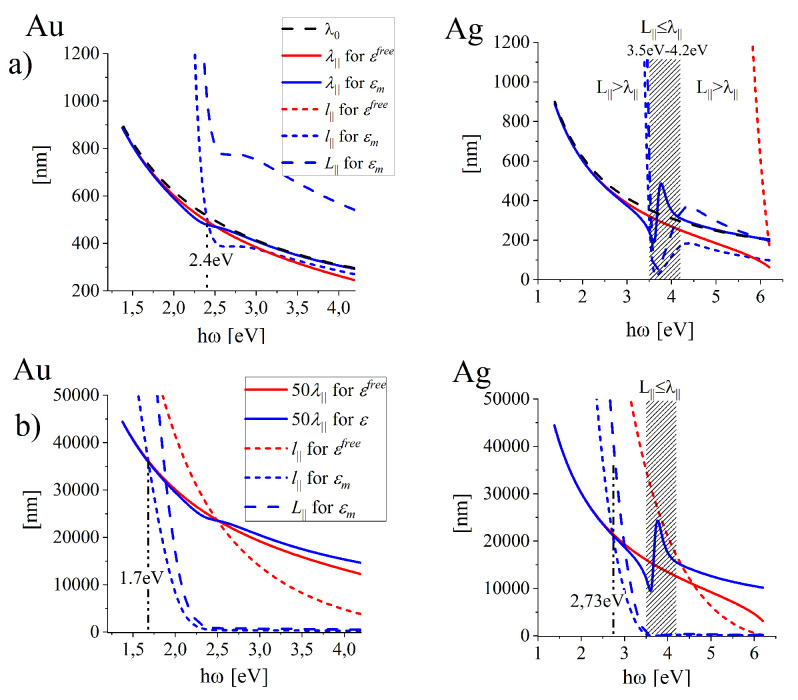
Illustration of the changes in predicted lengthscales of SPP for free-electron metals (ohmic losses included) and for the most reliable predictions (free-electron contribution also included) for (**a**) the SPP wavelength $\lambda_{\|}$, oscillation length $L_{\|}$ and the attenuation length $l_{\|}$ in gold and silver interfaces. (**b**) illustrate the corresponding length scales related to $50\lambda_{\|}$ which can be useful in the estimation of the frequency ranges for propagating SPP in applications.

**Figure 9 nanomaterials-10-01411-f009:**
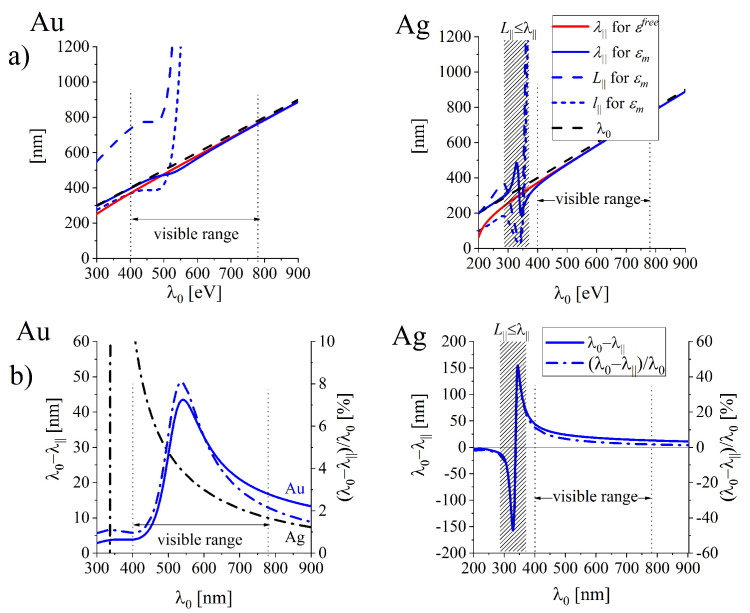
(**a**) SPP wavelength $\lambda_{\|}$, oscillation length $L_{\|}$ and the attenuation length $l_{\|}$ related to the wavelength $\lambda_{0}$ of the incoming field. (**b**) The absolute (solid lines) and relative (dot-dashed lines) shortening of the SPP wavelength of SPP wave at gold and silver interfaces.

**Figure 10 nanomaterials-10-01411-f010:**
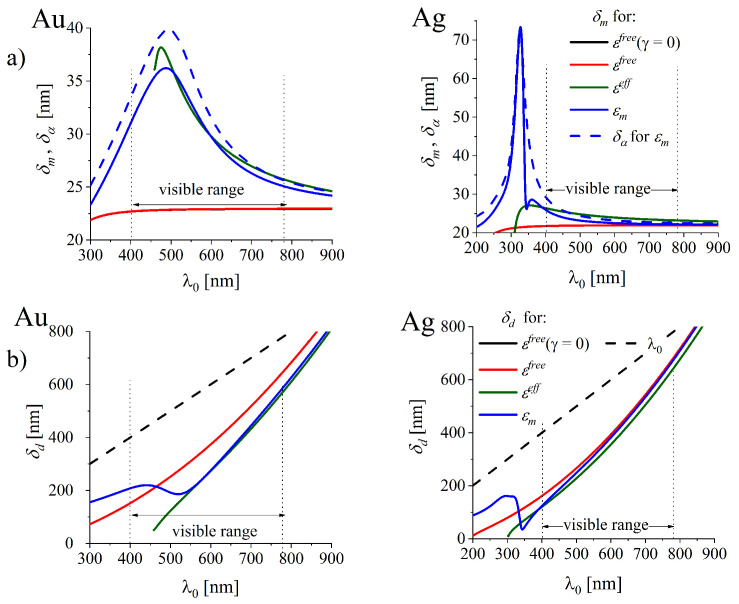
Confinement (localization) lengths $\delta_{m}$ and $\delta_{d}$ of SPP wave in the (**a**) metal and (**b**) dielectric in the direction perpendicular to the interface. Dashed blue lines in figs (**a**) illustrate the optical skin depth $\delta_{\alpha }$.

**Figure 11 nanomaterials-10-01411-f011:**
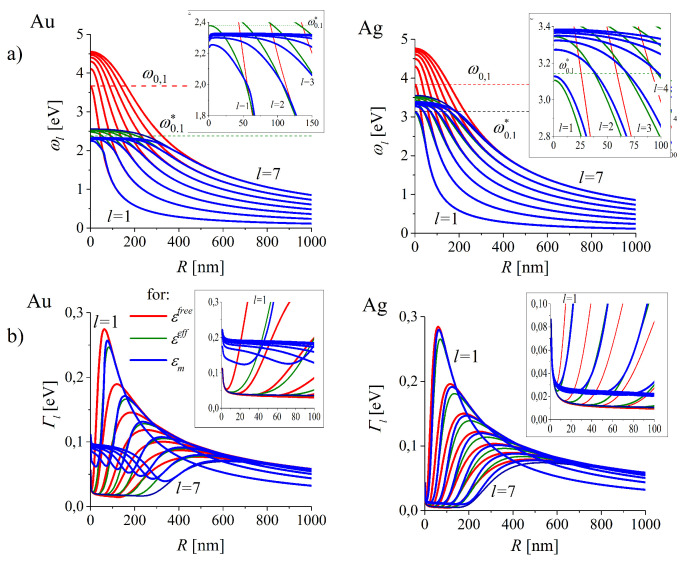
(**a**) Oscillation frequencies $\omega_{l}\left(R\right)$ and (**b**) damping rates $\Gamma_{l}\left(R\right)$ of LSP on gold (**left**) and silver (**right**) NPs for metals of properties represented by various models of the dielectric functions. Red, olive and blue lines represent the results for consecutive models of DFs: $\varepsilon^{free}\left(\omega ,R\right)$, $\varepsilon^{eff}\left(\omega ,R\right)$ and $\varepsilon_{m}\left(\omega ,R\right)$ correspondingly (Equations (Equation 1)–(Equation 3) adopted for MNPs) for gold (**left column**) and silver (**right column**).

**Figure 12 nanomaterials-10-01411-f012:**
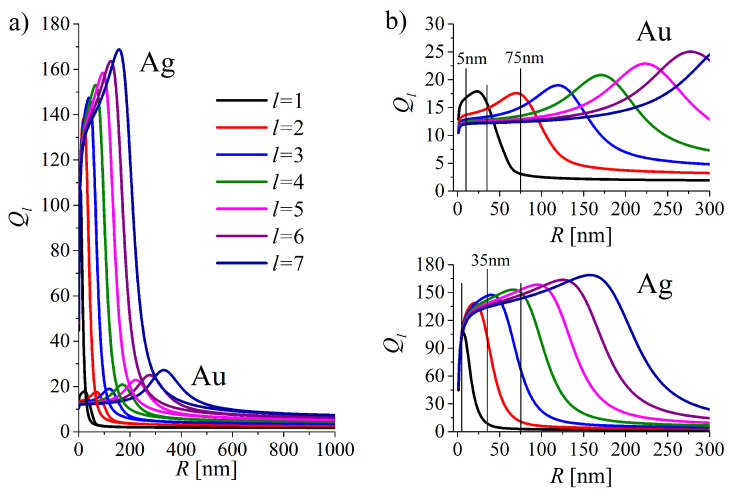
(**a**) Quality factors $Q_{l}$ for the modes *l* = 1,2,3...7 versus radius of gold and silver nanoparticles (NPs). Figure (**b**) demonstrate the zoomed results of figure (**a**) in order to highlight the range of sizes experimentally available.

**Figure 13 nanomaterials-10-01411-f013:**
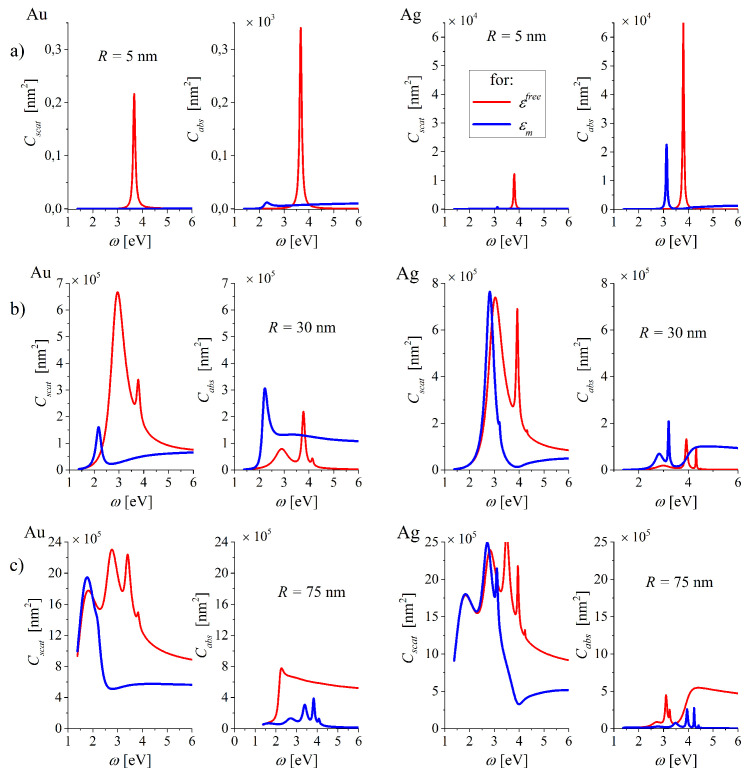
Scattering and absorption cross-sections for gold (left) and silver (right) nanospheres of radius (**a**) R=5 nm, (**b**) R=30 mn and (**c**) R=75 nm. Blue line for $\varepsilon_{m}$ and red line for $\varepsilon^{free}$.

**Figure 14 nanomaterials-10-01411-f014:**
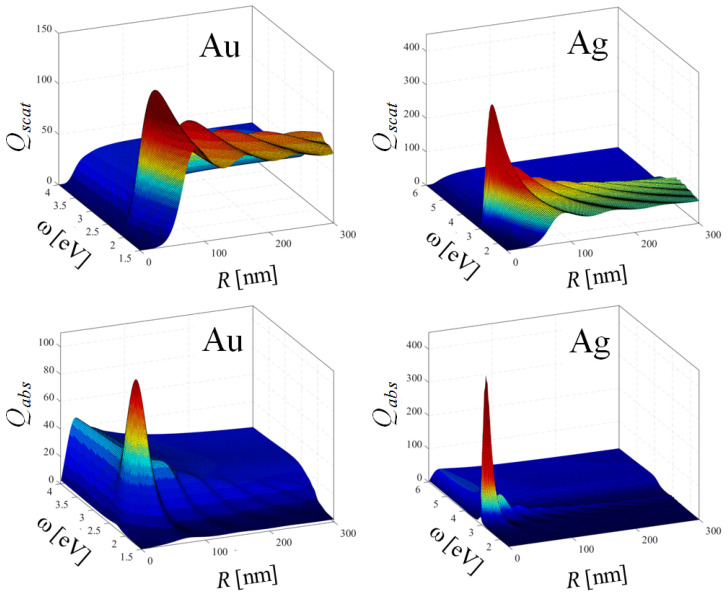
Scattering and absorption efficiencies $Q_{scat}\left(\omega ,R\right)=C_{scat}\left(\omega ,R\right)/\pi R^{2}$ and $Q_{abs}\left(\omega ,R\right)=C_{abs}\left(\omega ,R\right)/\pi R^{2}$ for gold (**left**) and silver (**right**) nanospheres for $n_{out}$ = 1.5.

**Table 1 nanomaterials-10-01411-t001:** External and the best-fitted parameters of the function $\varepsilon_{m}$ which describes the free-electrons contribution and the frequency-dependent contribution of the interband transitions in gold and silver.

$\varepsilon_{m}$	External	Fitted							Nr
	**Interband Contribution**						**Free electrons Contribution**		
	$\hbar \omega_{c}$ [eV]	$\varepsilon_{up}$	$\varepsilon_{lo}$	*S* [1/eV]	*A* [eV${}^{2}$]	${\left(\hbar \gamma_{L}\right)}^{2}$ [eV${}^{2}$]	$\omega_{p}$ [eV]	γ [eV]	
**Au**	2.4	5.6	-	5.87	7.78	0.977	8.6	0.072	7
**Ag**	3.9	6.37–0.54 [1/eV]·ℏω	0.1	7.90	4.02	1.00	9.0	0.020	8

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
