# Peer review of "Impact of the Interband Transitions in Gold and Silver on the Dynamics of Propagating and Localized Surface Plasmons"

_nanomaterials, 2020, doi:10.3390/nano10071411_

Round 1

Reviewer 1 Report

The topic and issues addressed in this manuscript are of relevance for the broad plasmonics field and community. The paper is solid, and it needs only minor language editing (some spelling/typing errors are present). 

Author Response

Thank you very much for your effort in reading the manuscript and the positive feedback.

Reviewer 2 Report

The authors are addressing an important issue that arises when modeling surface-plasmon phenomenon. When simplified models are used, it can lead to problems in modeling compared to experimental results. The authors claim that it is necessary to account for the radiative losses that are a result of frequency-dependent interband transitions.  The authors conducted a very thorough analysis for gold and silver nanoparticles. The authors demonstrated that accounting for the frequency-interband transition does have an impact on the propagating and localized surface plasmon properties and is better able to reproduces experimental data for both Au and Ag interfaces in extended frequency ranges.

Author Response

(The authors gave the same response as above.)

Reviewer 3 Report

In the manuscript entitled “Impact of the interband transitions in gold and silver on the dynamics of propagating and localized surface plasmons” the authors aim to present the shortcomings of Drude-type models for the dielectric function (DF) of gold and silver nanostructures and to correct them by including phenomenological interband transition terms. They treat both flat and nanoparticle metal/dielectric interfaces, i.e., surface plasmon polaritons (SPPs) and localized surface plasmons (LSPs). The manuscript contains a lot of information of potential interest to the metallic nanoparticle and plasmonics community and may published after some corrections and revisions which are mainly related to presentation.

In the introduction section, the problems with existing DF models should be clearly described with some specific citations from the long list of references of the paper, especially to experimental results. In section 3.3, more justification should be given for interband transition corrections, eq. (5) and (6), which are the central points of the paper. There is more discussion about eq. (5) in reference [43] but eq. (6) appears to be just a fit to experiment. Even if this is the case, “the simplest possible form of the Lorenz profile” (I would prefer “Lorentzian profile”) deserves a few more words. Similar to introduction, the conclusions section has to be more precise, highlighting results of this work related to specific experiments and/or predictions that are possible to check experimentally.

The presentation of the results is thorough and contains many figures that have to become larger, if the manuscript is published, making the publication (necessarily) longer. The authors need to correct minor errors, such as:

In section 2, page 3, gold atomic number is 79 (not 74).

In section 3.1, page 5, \omega_p and \gamma should be multiplied by \hbar.

In section 3.1.1, page 5, definition of “tau” may be obvious but not previously defined in the manuscript.

In Fig. 2, Au, b) inset “Re” should be “Im”.

Table I, page 7, Nr of parameters 6 should be 7.

Choose between “localization” and “localization” (both used in the text).

Other minor errors can be corrected, spell check will not detect some, such as “divers models”, “interbad”, “dumping”, “reviling”, “usage if simplified models”, etc.

Author Response

Thank you very much for your effort in reading the manuscript and valuable suggestions for corrections.

Below is the list of amendments made is attached, according to suggestions:

In the manuscript entitled “Impact of the interband transitions in gold and silver on the dynamics of propagating and localized surface plasmons” the authors aim to present the shortcomings of Drude-type models for the dielectric function (DF) of gold and silver nanostructures and to correct them by including phenomenological interband transition terms. They treat both flat and nanoparticle metal/dielectric interfaces, i.e., surface plasmon polaritons (SPPs) and localized surface plasmons (LSPs). The manuscript contains a lot of information of potential interest to the metallic nanoparticle and plasmonics community and may be published after some corrections and revisions which are mainly related to presentation.

In the introduction section, the problems with existing DF models should be clearly described with some specific citations from the long list of references of the paper, especially to experimental results.

  • page 3, Indroduction. The pargarph, which concern the models of the dielectric function have been supplemented by the text in italic:

    “The reliable dielectric function (DF) which describes the manner the EM field acts on electrons of the materials involved, is the input parameter of EM theory. Significance of realistic description of dielectric properties of noble metals in basic issues and applications has been a motivation to many experimental studies intended to increase the accuracy of measurements of the indexes of refraction of noble metals in the function of wavelength \cite{guerrisi1975splitting,johnson1972optical,ordal1985optical,palik1998handbook,blanchard2003high,web:ref.ind}. In studying many aspects of plasmonic problems, it is convenient to have at hand an analytic expression for the dielectric function $\varepsilon(\omega)$ applicable for real metals,  as it reveals the role of the useful parameters which characterize bulk metals. Such parameters are useful in predicting and shaping the plasmonic characteristics of the corresponding nanostructures. From this point of view, the realistic model of the frequency dependence of DF involving physically meaningful parameters and possibly small number of free parameters is of basic importance. The complex DF which reproduces well the measured real and imaginary parts of complex indexes of refraction of metals $n(\omega)$: $\varepsilon^{exp}(\omega)=n(\omega )^{2}=(n^{\prime }(\omega )+in^{\prime \prime }(\omega ))^{2}$ is a fundamental quantity which is directly related to the electronic structure of metals. Such DF  gives an insight into the elementary excitations of free carriers and interband absorption and allows us to study the basic surface plasmons characteristics supported by the interface. In particular, the model of DF for real metals must account for all important loss mechanism such as ohmic losses (heat release), inter-band absorption and scattering on surface.  The additional damping can be caused by the limited physical dimension of the system resulting in the restriction of the mean free path of electrons. The phenomenological expression for the rate of this process called a “surface collision scattering rate” \cite{kreibig1995optical} is proportional to $v_F/a$ where $v_F$ is the Fermi velocity and $a$ is the characteristic dimension of the metallic object.  Let us note that such additional damping with the phenomenological “surface collision scattering rate” in nanoscale metals can be connected with Landau damping \cite{khurgin2015ultimate, khurgin2017landau} and quantum confinement effect which takes place near metal-dielectric interfaces. However, in many studies, DF is modeled using a Drude-Lorentz approach which accounts for the intraband behavior of metals only. Such models, as well as those which approximately account for the presence of the interband transitions by a constant added to the real part of DF (e.g. \cite{maier2007plasmonics,derkachova2016dielectric}), are known to be unsatisfactory in reproducing the experimentally measured indexes of refraction of noble metals.  The frequency dependence of the interband transitions of metals is usually modeled by one or more Lorentzian-shaped contributions to DF what requires the use of large number parameters \cite{kreibig1995optical,etchegoin2006analytic,hao2007efficient,rioux2014analytic}. The optical properties of gold are more difficult to represent in the visible and near-UV region with an analytic model due to the more important role of interband transitions in these regions (e.g \cite{kreibig1995optical}). .”

In section 3.3, more justification should be given for interband transition corrections, eq. (5) and (6), which are the central points of the paper. There is more discussion about eq. (5) in reference [43] but  eq. (6) appears to be just a fit to experiment. Even if this is the case, “the simplest possible form of the Lorenz profile” (I would prefer “Lorentzian profile”) deserves a few more words.

  • page 10, the text concerning eq.(5) have been changed to:
    We introduce such frequency-dependent correction in harmonic-oscillator-based representations of  the  dielectric  functions in the simplest possible form of the Lorentzian profile:

\begin{equation}

\operatorname{Re}\varepsilon^{ib}(\hbar\omega) =

\frac{A} {\hbar^2 (\omega - \omega_c)^2 + (\hbar\gamma_L)^2},

\end{equation}\label{ReEpsilon^ib}

with two fitted parameters $A$ and $\gamma_L^2$. The many-parameter Lorentzian profiles have been previously used in the DFs models basing e.g. on critical points analysis of metal band-structure (e.g.\cite{etchegoin2006analytic,rioux2014analytic}).

  • page 10, the Lorenz profile have been replaced by the Loretzian profile

Similar to introduction, the conclusions section has to be more precise, highlighting results of this work related to specific experiments and/or predictions that are possible to check experimentally.

  • page 24: Section 4.2.1. Paragraph “Inclusion of interband transitions...”
    have been replaced by:
    “Inclusion of interband transitions (accounted in $\varepsilon^{eff} (\omega, R) $ and $\varepsilon_m (\omega, R)$) possesses an essential impact on $\ omega_l(R $ with the decreasing $R$. The experimentally measured resonance positions \cite{derkachova2016dielectric,sonnichsen2001plasmons,khlebtsov2010optical,njoki2007size} (which are usually limited of the dipole ($l=$1) plasmon resonance only) are more accurately predicted by a modeling using $\varepsilon_m(\omega)$. For example, in the experiment using gold nanoparticles with slight deviations from a perfectly spherical shape embedded in oil ($n_out=$1.5) \cite {sonnichsen2001plasmons}, the resonance positions are 2.18eV (568nm ) for the radius $ R = 30 $ nm , what coincide very well with our predictions (see the insets in Figures \ref{fig:DR_NPs} a)). For gold and silver nanoparticles in water ($n_out = $1.3) the predictions basing on $\varepsilon_m (\omega, R)$ (not presented here) also fit well to the experimental results \cite{derkachova2016dielectric, sonnichsen2001plasmons, khlebtsov2010optical, njoki2007size}.”
  • page 30 in Conclusions section: The fragment which have been modified:
    In particular, consideration of the presence of frequency-dependent transitions in the extended frequency range revealed the presence of anomalous dispersion behavior for the silver flat interface in the near-UV, which, as far as we know, has not been experimentally studied.
  • page 31 in Conclusions section: added:
    Therefore, among many experimental studies which deal with SPPs in thin films, hybrid structures and nanowires (e.g. \cite{schider2003plasmon,kostiuvcenko2014surface,kawalec2018surface} ), the experimentally obtained dispersion relation bends to the higher frequencies ("to the right of the dispersion curve"), if compared with predictions based on simplified models of DFs.  In particular, in a case of nanowires  (e.g. \cite{schider2003plasmon}, the conclusion of slowing down the propagation velocity of SPP compared to the corresponding planar interface is not fully justified as the bending of the experimental dispersion curve can result from the presence of interband transitions which are not accounted in the modeling.

Inclusion of interband transitions (accounted in $\varepsilon^{eff} (\omega, R) $ and $\varepsilon_m (\omega, R)$) possesses also an essential impact on predicted self-energies $\omega_l(R)$ of spherical metal interfaces. The experimentally measured spectral positions of LSPs manifestation \cite{derkachova2016dielectric,sonnichsen2001plasmons,khlebtsov2010optical,njoki2007size} (which are usually limited to the dipole plasmon resonance ($l=$1) only) are very well  predicted by the presented  modeling using $\varepsilon_m(\omega)$ for $n_out=$1.5 (the case studied above) and, as we checked also for $n_out=$1.3  (i.e. for MNPs embedded in the immersing oil and water correspondingly).

The presentation of the results is thorough and contains many figures that have to become larger, if the manuscript is published, making the publication (necessarily) longer. The authors need to correct minor errors, such as:

In section 2, page 3, gold atomic number is 79 (not 74).

In section 3.1, page 5, \omega_p and \gamma should be multiplied by \hbar.

In section 3.1.1, page 5, definition of “tau” may be obvious but not previously defined in the manuscript.

In Fig. 2, Au, b) inset “Re” should be “Im”.

Table I, page 7, Nr of parameters 6 should be 7.

Choose between “localization” and “localization” (both used in the text).

Other minor errors can be corrected, spell check will not detect some, such as “divers models”, “interbad”, “dumping”, “reviling”, “usage if simplified models”, etc.

  • page 3: the atomic number of gold have been changed to 79
  • page 5: Fig.1 caption: n(\omega) changed by n(\hbar\omega)
  • page 7
    (top):
    • ($\omega_p=$8.6eV and $\gamma=$0.072eV) and silver ($\omega_p=$9.0eV and $\gamma=$0.02eV)
      have been replaced by:
      ($)\hbar \omega_p=$8.6eV and $)\hbar \gamma=$0.072eV) and silver ($)\hbar \omega_p=$9.0eV and $)\hbar \gamma=$0.02eV
    • $\operatorname{Im}\varepsilon^{exp}(\omega)$ and $n'(\omega)$ and about 2eV for $\operatorname{Re}\varepsilon^{exp}(\omega)$ and $n"(\omega)$ )
      have been replaced by:
      $\operatorname{Im}\varepsilon^{exp}(\hbar\omega)$ and $n'(\hbar\omega)$ and about 2eV for $\operatorname{Re}\varepsilon^{exp}(\hbar\omega)$ and $n"(\hbar\omega)$

 (bottom):

  • $\operatorname{Im}\varepsilon^{exp}(\omega)$ (e.g., those using $\tau=(9,3\pm 0.9)$s)
    have been replaced by:
    $\operatorname{Im}\varepsilon^{exp}(\hbar\omega)$ (e.g., those using the relaxation time $\tau=(9,3\pm 0.9)$fs ($\tau=1/\gamma$)
  • Page 8: Accepted parameters for gold allow good reproduction of $\operatorname{Re}\varepsilon^{exp}(\omega)$ and $n"(\omega)$ in the frequency range up to about 2.9eV and $\operatorname{Im}\varepsilon^{exp}(\omega)$ and $n'(\omega)$ up to about 1.8eV
    have been replaced by:
    Accepted parameters for gold allow good reproduction of $\operatorname{Re}\varepsilon^{exp}(\hbar\omega)$ and $n"(\hbar\omega)$ in the range up to about 2.9eV and  $\operatorname{Im}\varepsilon^{exp}(\hbar\omega)$ and $n'(\hbar\omega)$ up to about 1.8eV
  • In Fig. 2, Au, b) inset “Re” have been replaced by “Im”.
  • Table I, Nr of parameters have been changed from 6 to 7.
  • “localisation” have been changed by “localization”
  • “divers models” have been replaced by “various models”
  • “interbad” have been replaced by “interband”
  • “dumping” have been replaced by “damping”
  • “reviling” have been replaced by “revealing”
  • “usage if simplified models” have been replaced by  “usage of simplified models”,

Reviewer 4 Report

The manuscript in my opinion is well written and can be considered for the publication on Nanomaterials.

Author Response

Thank you very much for your effort in reading the manuscript and your positive opinion.

Below is the list of amendments made is attached, according to suggestions:

  • “localisation” have been changed by “localization”
  • “divers models” have been replaced by “various models”
  • “interbad” have been replaced by “interband”
  • “dumping” have been replaced by “damping”
  • “reviling” have been replaced by “revealing”
  • “usage if simplified models” have been replaced by  “usage of simplified models”,